# Patterns of smartphone typing performance by time awake: implications for unobtrusive ambulatory mental fatigue assessment

Yu Fang[1]*, Peter Yang[2], Elena Frank[1], Cathy Goldstein[3], Aidan G. C. Wright[4,5], Amy S. B. Bohnert[6,7], Vik Kheterpal[8], Srijan Sen[1,5,6], Zhenke Wu[9]

1 Michigan Neuroscience Institute, University of Michigan Medical School, Ann Arbor, Michigan, United States of America, 2 Department of Statistics, University of Michigan, Ann Arbor, Michigan, United States of America, 3 Department of Neurology, University of Michigan Medical School, Ann Arbor, Michigan, United States of America, 4 Department of Psychology, University of Michigan, Ann Arbor, Michigan, United States of America, 5 Eisenberg Family Depression Center, University of Michigan, Ann Arbor, Michigan, United States of America, 6 Department of Psychiatry, University of Michigan Medical School, Ann Arbor, Michigan, United States of America, 7 Department of Anesthesiology, University of Michigan Medical School, Ann Arbor, Michigan, United States of America, 8 CareEvolution, Ann Arbor, Michigan, United States of America, 9 Department of Biostatistics, University of Michigan, Ann Arbor, Michigan, United States of America

* yfang@med.umich.edu

## Abstract

Mental fatigue undermines workplace safety and productivity. Early detection of subtle declines in objective alertness and cognitive performance in workers can enable timely interventions to prevent costly errors and safeguard employee health. However, conventional assessments often require controlled laboratory conditions and prolonged testing, limiting their real-world applicability. Here, we demonstrate an approach that utilizes smartphone keyboard metrics in everyday use to provide a more scalable, continuous, and ambulatory method for evaluating mental fatigue. We examined the adjusted association between novel yet widely available SensorKit typing performance metrics and wearable-derived time since waking from a most recent major sleep episode or napping among 366 first-year training physicians in the United States who generated 45,042 typing sessions over a two-month period. Typing performance, especially typing speed, has a significant non-linear adjusted relationship with time awake. At the population-level, typing speed increases and peaks around 7.5 hours since awake and has a substantial decrease around 15.3 hours of time awake, highly consistent with classical lab-based active task Psychomotor Vigilance Test findings that showed increased response lapses beyond 15.8 hours of wake period. Our findings are relevant for developing a ubiquitous and unobtrusive tool to assess, monitor, and manage mental fatigue on a continuous basis in everyday life, especially for populations in high-risk and high-stake settings.

**Data availability statement:** SensorKit data cannot be shared publicly because of Apple's data sharing restrictions. De-identified general study enrollment and survey data are available via the ICPSR repository (https://www.open-icpsr.org/openicpsr/project/129225/version/V1/view). De-identified wearable data is available through requests to the Intern Health Study (intern_health@med.umich.edu). Reasonable requests will be fulfilled contingent on completion of a data use agreement with the University of Michigan.

**Funding:** This work was supported by the National Institutes of Mental Health (R01MH101459 to SS, ZW). The funders had no role in study design, data collection and analysis, decision to publish, or preparation of the manuscript. No authors received any salary from any of the funders.

**Competing interests:** The authors have declared that no competing interests exist.

## Author summary

Mental fatigue can impact people's safety and performance at work, especially in demanding jobs like medicine or shift work. We wanted to find an easy and effective way to measure when people start feeling mentally tired, without interrupting their regular work and life. In this study, we followed 366 new doctors across the United States as they typed on their iPhones as they did their usual daily activities. By collecting data from over 45,000 typing sessions through the native keyboard sensor, and combining it with information about their sleep, we noticed clear changes in how fast people typed as they became more tired. Typing speed increased over the first several hours after waking up, but then showed a clear slowdown as people stayed awake longer, closely matching results from special attention tests done in special labs. Our work shows that everyday smartphone typing could reveal early signs of mental fatigue in real time. This approach could help people in high-stress jobs track their own alertness and take steps to prevent mistakes, ultimately supporting safer and healthier workplaces.

## Introduction

Mental fatigue is a driving risk factor for impairment of workplace safety and productivity and increase in work errors, especially for occupations with long work hours or shift schedules, such as healthcare workers [1, 2]. Mental fatigue in real-world performance is predominantly influenced by two main factors: sleep deprivation and prolonged time on task. At present, the methods to reliably measure mental fatigue require active participation. For instance, the Psychomotor Vigilance Test (PVT), the most widely utilized tool for detecting these fatigue-related signals [3], requires several minutes of active participation to complete, even with the developed brief versions [4], and is generally restricted to laboratory settings. Thus, current tools cannot be used for real-time, continuous detection of mental fatigue signals in the real world without interrupting regular activities. Therefore, alternative measures are needed to address the gap.

Recent research has investigated the connection between typing performance and mental fatigue, using both naturalistic and controlled environments to assess how cognitive functions influence typing speed and accuracy. Pimenta et al. proposed an unobtrusive method for measuring an individual's level of fatigue through the interaction patterns of the individual with the mouse and keyboard of the computer [5, 6]. Al-libawy et al. proposed various methods to predict fatigue status using smartphone text entry performance metrics such as hold time and digraph time [7, 8]. de Jong et al. further reported that typing markers of computer keyboard could inform the mental fatigue status, by showing that decreased P3 brain potential amplitude (known to be related to mental fatigue [9–11]) recorded by electroencephalography (EEG) is related to increase in typing time and typing errors [12]. Ross et al. discovered that typing speed on smartphones is positively associated with performances in the digital version of trail making test, a neuropsychological test that assesses visual attention,

processing speed, and task switching [13]. Based on these validations of using typing performance as an indicator of mental fatigue, de Jong et al. found that typing markers are susceptible to changes in behavior related to mental fatigue, with dynamic changes over different time scales such as time-on-task, time-of-day, and day-of-week in an uncontrolled work environment [14]. A study of emergency medicine resident physicians demonstrated significant decrements in typing speed during night shift compared to day shift and evening shift [15]. These studies underscore the value of typing performance as a non-intrusive and reliable indicator for monitoring mental fatigue across various settings. However, these studies have been constrained by small sample sizes (all fewer than 100) and the need for a customized keyboard application to capture typing metrics, limiting the robustness of their conclusions and the utility and scalability of the tools utilized.

SensorKit (Apple Inc.) [16,17] is an innovative technological framework on iPhone that enables researchers to request relevant sensor data from consented participants via a study app that reads the data on-device after said participant grants permission. Metrics from on-device sensors for iPhone help to optimize user experience without sharing personal data off-device, including not with Apple. Specifically, the keyboard sensor captures the configuration of the phone's keyboard and monitors its usage pattern, such as typing speed and accuracy, during the use of any App or other phone function, without recording the actual content typed or identifying the App or function. This capability provides a powerful tool for passively examining typing performance on a large scale, without the need for additional software installation. With over 1.3 billion active iPhone users worldwide [18], the scalability of a passive iPhone based mental fatigue marker is immense. This study aims to utilize the data collection capabilities for the native iPhone keyboard to investigate whether passively collected keyboard performance could serve as an indicator of mental fatigue. To accomplish this aim, this study considers first year medical residents (interns), who typically endure long work hours, difficult decisions, and sleep deprivation, representing an ideal population for studying mental fatigue.

## Methods

### Ethics statement

The study was approved by the University of Michigan Institutional Review Board (HUM00033029) and all subjects provided informed consent after receiving a complete description of the study.

### Study design

The Intern Health Study is an annual cohort study that follows the stress and mental health of first-year physician residents (interns) across the United States. In the 2023 cohort of Intern Health Study, interns starting residency in 2023 were invited to participate via email 2–3 months before the start of internship (July 2023). Participants completed the main study consent and a baseline survey that collected demographic information and then logged into the MyDataHelps mobile application (CareEvolution, Ann Arbor, MI). Upon enrollment, participants were provided with their choice of wearable devices including Fitbit Charge 4, Inspire 2 (Google Fitbit, San Francisco, CA), Apple Watch (Apple Inc., Cupertino, CA), or $50 if they already have a Fitbit, Apple watch or Garmin watch (Garmin Ltd., Olathe, KS). Interns received up to $80 additional compensation throughout the year for continued participation. iPhone users among the participants were offered enrollment in the SensorKit arm of the study (Ambient Light, Keyboard Usage, Message Usage, Phone Usage, and Visits) through additional consenting processes, see Funk et al. 2025 for details [19]. No additional incentives were provided for opting into the collection of SensorKit data.

### Data collection

**iPhone keyboard metrics.** The keyboard usage and measurement of participants were collected through SensorKit, which is a novel framework supported by Apple allowing researchers to obtain additional data collected from sensors of consented study participants' iPhones. The Keyboard Usage sensor [20] of SensorKit provides information about keyboard usage including typing duration (seconds, '*totalTypingDuration*'), typing speed (characters per second,

'*typingSpeed*'), total taps (counts, '*totalTaps*'), and total deletes (counts, '*totalDeletes*'). Other keyboard features, such as width and height of the keyboard (both in millimeters), as well as the active keyboard languages ('*inputModes*') were also recorded. The date and time of each typing event were recorded, and the typing events were grouped to the nearest half-hour session for privacy reasons (i.e., daily at 00:00, 00:30, …, 23:00, 23:30, 24:00). For privacy protection, any of the words typed, or any information about keyboard extensions or stickers were not collected (S1 Fig).

The iPhone keyboard metrics from the first two months (July and August, 2023) of the intern year were included in the analysis. From the original measures of keyboard usage, we derived two metrics that reflect average typing performance for the multiple typing events (*i*) grouped by typing session at each day's each half-hour grid point (*t*), where t represents the actual date and time stamp rounded to the nearest half-hour. All subjects share the same set of grid points; we calculate the typing metrics for subject *s* at *t* only if there is at least one associated typing session. First, the average typing speed for subject *s* at *t* was calculated by:

$$\text{TypingSpeed}_s(t) = \frac{\sum_i \left\{ \text{totalTypingDuration}_s(t, i) \times \text{typingSpeed}_s(t, i) \right\}}{\sum_i \text{totalTypingDuration}_s(t, i)}$$

Second, the average typing accuracy, represented by the rate of deletions, for subject *s* at *t* was calculated by:

$$\text{DeleteRate}_s(t) = \frac{\sum_i \text{totalDeletes}_s(t, i)}{\sum_i \text{totalTaps}_s(t, i)}$$

Finally, because this study focuses on examining changes within individuals over the course of time since awake, typing speed and rate of deletions were converted to standardized z-scores for each subject *s* at *t*:

$$\text{TypingSpeed Zscore}_s(t) = \frac{\text{TypingSpeed}_s(t) - \mu\left(\text{AllRecordedTypingSpeed}_s\right)}{\sigma\left(\text{AllRecordedTypingSpeed}_s\right)}$$

and

$$\text{DeleteRate Zscore}_s(t) = \frac{\text{DeleteRate}_s(t) - \mu\left(\text{AllRecordedDeleteRate}_s\right)}{\sigma\left(\text{AllRecordedDeleteRate}_s\right)}$$

where $\mu(\cdot)$ and $\sigma(\cdot)$ represent the mean and the standard deviation of all records of typing speeds ($\text{AllRecordedTypingSpeed}_s = \left\{ \text{TypingSpeed}_s(t) : \text{if subject s typed at t} \right\}$) or deletes rate ($\text{AllRecordedDeleteRate}_s = \left\{ \text{DeleteRate}_s(t) : \text{if subject s typed at t} \right\}$) over all day all time points for subject *s,* respectively.

## Sleep measures

Similar to validation tests for the PVT [21], we used time awake as the metric representative of the level of mental fatigue due to the build-up of the homeostatic sleep drive during sustained wakefulness. Sleep timing (bedtime and wake time) of the participants were recorded by the wearable devices. Fitbit and Garmin watches directly provided the bedtime and wake time of each sleep episode. Apple Watch provided intraday sleep stage data via HealthKit [22], from which we combined any sleep stages that were separated by gaps of <30 mins into one sleep episode for each subject, and extracted the bedtime and wake time from each sleep episode.

Next, for each subject, we linked each typing session with a preceding sleep episode if 1) no other sleep episode occurred in between; 2) the time interval between the previous wake time and the typing session was no more than 24 hours. Then for each typing session, time awake was derived as the time interval between the timestamp of the typing initiation (rounded to the nearest half hour) and the wake time of its linked most recent sleep episode.

## Statistical analysis

All statistical analyses were conducted with the use of R version 4.3.2 (The R Foundation, Vienna, AUT) [23].

We utilized generalized estimating equations (GEE) modeling, which is capable of adjusting for repeated measures from the same subject, to formally examine the population-average association between time awake and typing performance. We used an independent working correlation matrix to produce robust inferential results to protect against potential endogeneity [24]; however, this analysis still acknowledges the dependence of typing performance outcomes within each subject in the calculation of the coefficient estimates, intervals, and p-values. The analyses were performed with R packages 'geepack' [25] and 'glmtoolbox' [26].

We fitted two sets of models, one with typing speed zscore as the outcome, and the other with rate of deletion zscore as the outcome. To account for potential non-linear relationships, both linear and quadratic terms of time awake were included as primary predictors in the analysis. Three nested models were analyzed:

$$\mathbb{E}[\text{Typing performance}] = \beta_0 + \beta_1(\text{time awake}) + \beta_2(\text{time awake})^2, \tag{1}$$

$$\mathbb{E}[\text{Typing performance}] = \beta_0 + \beta_1(\text{time awake}) + \beta_2(\text{timeawake})^2 + \gamma \text{covariates}, \tag{2}$$

$$
\begin{aligned}
\mathbb{E}[\text{Typing performance}] = {} & \beta_0 + \beta_1(\text{time awake}) + \beta_2(\text{time awake})^2 \\
& + \beta_3(\text{time awake} \times \text{age}) + \beta_4(\text{time awake} \times \text{sex}) \\
& + \beta_5(\text{time awake} \times \text{wearable device types}) + \beta_6(\text{time awake} \times \text{last sleep duration}) \\
& + \beta_7\left(\left(\text{timeawake}\right)^2 \times \text{age}\right) + \beta_8\left(\left(\text{time awake}\right)^2 \times \text{sex}\right) \\
& + \beta_9\left(\left(\text{time awake}\right)^2 \times \text{wearable device types}\right) \\
& + \beta_{10}\left(\left(\text{time awake}\right)^2 \times \text{lastsleep duration}\right) \\
& + \gamma \text{ covariates.}
\end{aligned} \tag{3}
$$

The first model assessed the relationship between the typing performance metrics and time awake without adjustment for other covariates. The second model examined this relationship after adjusting for covariates, including age, biological sex, time of day (modeled using the sine and cosine of its radian format to capture the circular nature of time and flexibly represent rhythms of any phase and amplitude), total typing duration (and its quadratic term), duration of previous sleep period (and its quadratic term), whether the day is weekend, wearable device types, keyboard area, and the other typing performance metric (if the outcome is typing speed zscore, then rate of deletion zscore is included as a covariate, vice versa). The third model further explored whether factors including age, sex, wearable device types, and duration of last sleep episode moderated the relationship between typing performances and time awake. Model fit was compared by the Schwartz-type penalized Gaussian Pseudo-likelihood Criterion (SGPC) (lower values are better) which was shown to outperform other existing methods in selecting variables and performs well regardless of whether the working correlation structure is correctly specified or not [27,28].

When visualizing the smoothed relationship between time awake and typing performances, we included a generalized additive model (GAM) smoothed prediction fitting line using a model similar to model (2), but replacing the linear and quadratic terms of time awake with a natural spline term (df = 3) of time awake, to better capture the possible higher order of non-linear relationship:

$$
\begin{aligned}
\mathbb{E}[\text{Typing performance}] = {} & \beta_0 + \beta_1 \, \text{natural\_spline}(\text{time awake}, \text{df} = 3)_1 \\
& + \beta_2 \, \text{natural\_spline}(\text{time awake}, \text{df} = 3)_2 \\
& + \beta_3 \text{natural\_spline}(\text{time awake}, \text{df} = 3)_3 \\
& + \gamma \text{ covariates.}
\end{aligned} \tag{4}
$$

PLOS Digital Health

The typing performances could potentially also be affected by time of day, independent of hours awake. To test whether typing performances present diurnal rhythmicity, we fit cosinor models, with typing performances as outcome, and time of day as the time variable, using the 'cosinor' package.[29] In addition, to separate the effects of time awake and time of day on typing performances, we made an illustration with 4 strata where time awake was set at less than 6 hours, 6–12 hours, 12–18 hours, and 18–24 hours. Each stratum was fit with model (4) respectively.

### Sensitivity analyses

First, to mitigate potential skewness arising from a small number of typing sessions occurring beyond 18 hours awake, we repeated the analysis by restricting to the typing sessions with a maximum of 18 hours awake.

Second, to eliminate the impact of different language settings on typing performance, we repeated the analysis by restricting to the typing sessions where the input mode was US English ('en_US').

Third, to clarify whether wearable device type affected results, we conducted a sensitivity analysis by repeating the same analysis within three subgroups that contain typing sessions associated with only Fitbit-recorded, Apple Watch-recorded, or Garmin-recorded sleep episodes. Each subgroup was fit with its own model:

$$\mathbb{E}[ \text{Typing performance} ] = \beta_0 + \beta_1 \text{ natural\_spline ( time awake, df = 3)}_1$$
$$+ \beta_2 \text{ natural\_spline ( time awake, df = 3)}_2$$
$$+ \beta_3 \text{ natural\_spline ( time awake, df = 3)}_3$$
$$+ \gamma \text{ covariates (excluding types of wearable devices).} \tag{5}$$

Finally, the wearable and SensorKit data analyzed were collected from the first two months of medical internship, however, it is important to note that sleep at baseline (the period after enrollment but before the beginning of internship) might have influenced the results. To account for this, we re-ran model (3) including only the participants with baseline sleep data and added baseline sleep duration as an additional covariate, both with itself and its interaction with time awake.

## Results

### Sample characteristics

4,151 interns starting residency in July 2023 were invited and 1,437 interns enrolled in 2023 Intern Health Study. After excluding non-iPhone users, subjects who opted out of the SensorKit keyboard usage data collection, and the ones who didn't provide sufficient data, 366 participants from 24 specialties and 154 residency programs were included in the final analyses (see **Fig 1** for study flow chart). **Table 1** provides the descriptive statistics of demographics and the collected typing and sleep data for the included subjects. There were 212 (57.9%) female participants, with mean age of 27.7 years old (SD = 2.7). The proportion of Fitbit, Apple Watch, and Garmin Watch users were 15.0%, 80.1% and 6.3%, respectively, with 5 subjects using more than one type of device (not on the same day). In all, 45,042 typing sessions linked with a preceding sleep episode were recorded. The individual mean typing speeds ranged from 2.1 to 6.6 characters per second (c/s) and averaged at 4.6 c/s (SD = 0.8 c/s). The individual mean rate of deletion ranged from 1% to 25%, and averaged at 7.3% (SD = 3.0%). The average of individual median wake up time was at 06:23 am (SD = 108.7 minutes). Time awake was significantly correlated with both the sine (r = -0.46, p < 0.001) and cosine (r = 0.67, p < 0.001) transformations of time-of-day, i.e., the time awake tends to be longest near midnight (peak of cosine) and shortest in the morning (peak of sine). S2 Fig shows the distributions of time awake, typing speed, and rate of deletion of the recorded typing sessions. **Fig 2** illustrates an example of recorded sleep episodes and typing sessions of one random subject throughout 62 days.

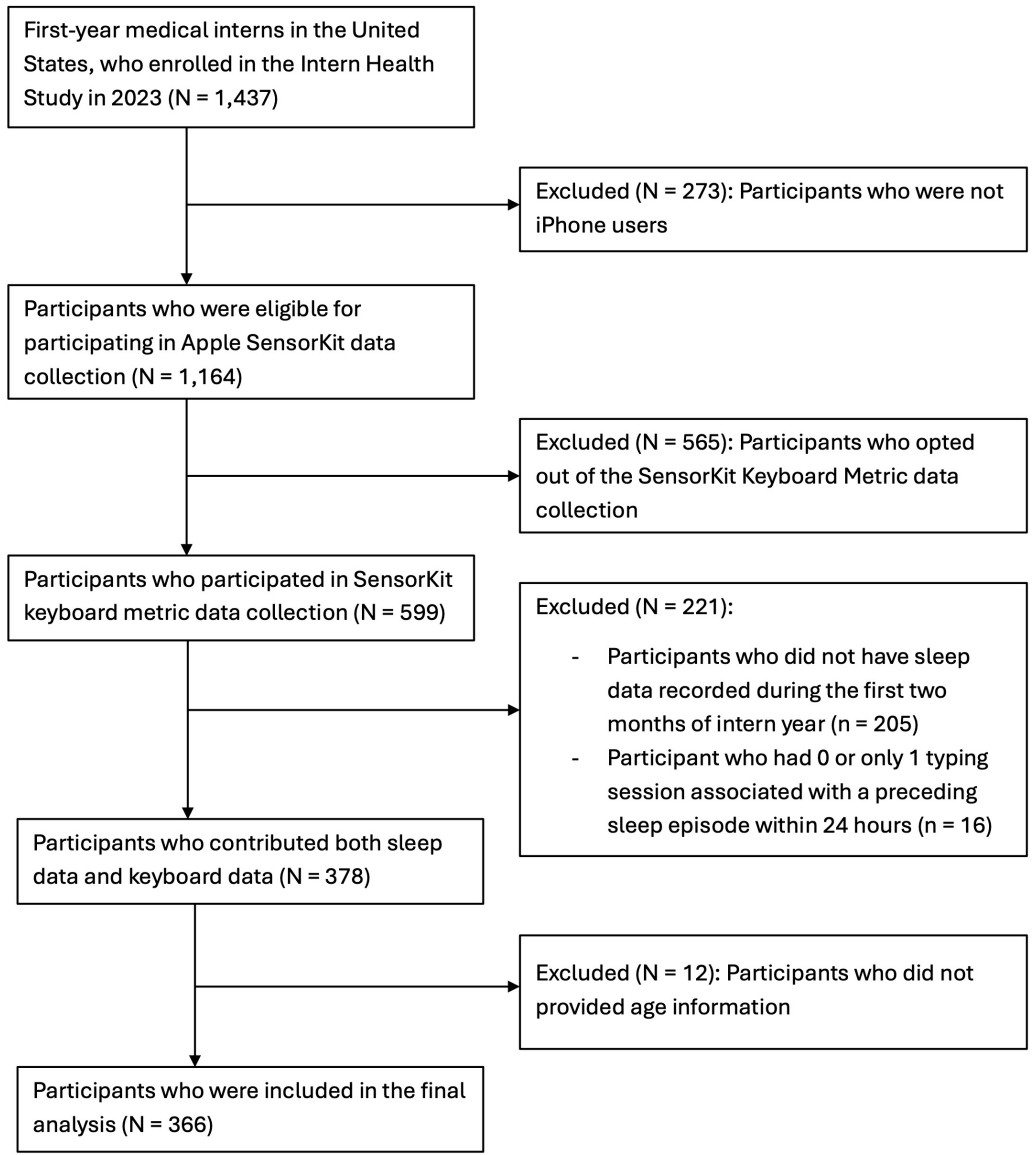

**Fig 1. Study sample flow chart.**

### The relationship between time awake and typing speed

According to SGPC, the adjusted model (2) had the best fit for the relationship between time awake and typing speed. Based on model (2), typing speed was significantly associated with both the linear and quadratic terms of time awake, after adjusting for the covariates (**Table 2, middle column**). The negative coefficient for quadratic term of time awake suggests that the relationship between time awake and typing speed follows an inverted U-shape, as shown in **Fig 3A**. Based on the GAM-smoothed prediction from model (4), the typing speed increased upon waking and peaked to approximately 0.098 SD above average at around 7.5 hours after wake-up, and declined to over 0.1 SD below average at about 15.3 hours awake. The numbers of typing sessions occurring after being awake for more than 19 hours were relatively lower (S2A Fig) thus the error bars were wider for these times.

**Table 1. Sample characteristics (N = 366).**

| Sample characteristics | |
|---|---|
| Age (years), mean (SD) | 27.7 (2.7) |
| Sex (female), N (%) | 212 (57.9) |
| Wearable Device Type, N (%) | |
| Apple Watch | 293 (80.1) |
| Fitbit | 55 (15.0) |
| Garmin | 23 (6.3) |
| Number of Typing Sessions of each Subject, mean (SD) | 123 (139) |
| Duration of Typing Sessions (seconds), mean (SD) | 141 (128) |
| Number of Sleep Episodes of each Subject, mean (SD) | 28 (19) |
| Duration of Sleep Episodes (minutes), mean (SD) | 417 (103); 6.95 hrs (1.72hrs) |
| Individual mean Typing Speed (characters per second), mean (SD) | 4.6 (0.8) |
| Individual mean Rate of Deletion (%), mean (SD) | 7.3 (3.0) |
| Individual median wake up time, mean (SD) | 06:23 am (108.7 mins) |

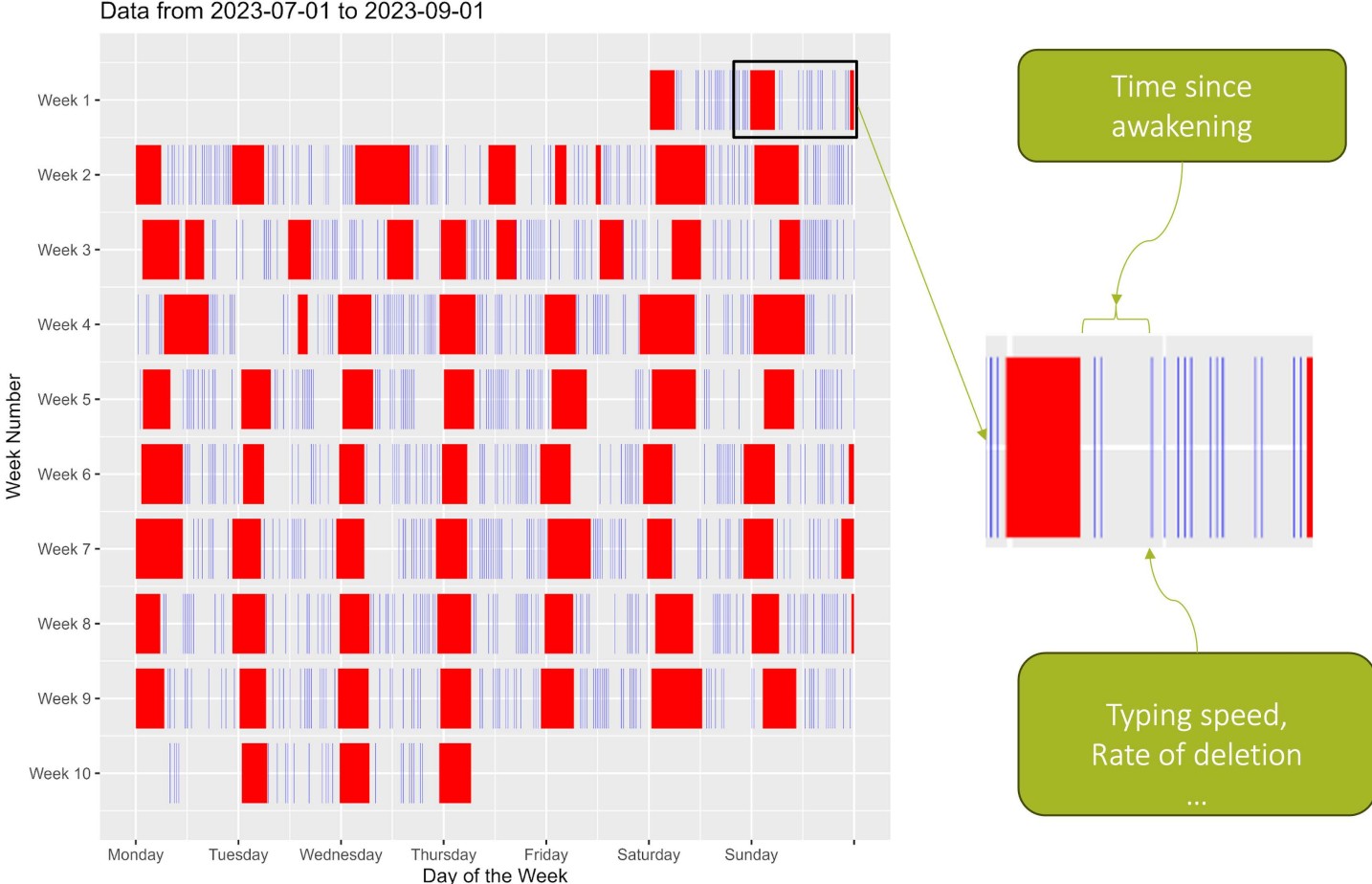

**Fig 2. Recorded sleep episodes (red blocks) and typing sessions (blue bars) for a randomly chosen subject between the start of internship on July 1st to September 1st, 2023.** Recall that each typing session was rounded to the nearest 30-min mark for privacy considerations and may contain multiple typing events.

Table 2. The relationship between time awake and typing speed. Left: unadjusted - model (1); Middle: adjusted - model (2); Right: adjusted with interactions - model (3).

| Predictors | Typing speed Zscore | | | Typing speed Zscore | | | Typing speed Zscore | | |
|---|---|---|---|---|---|---|---|---|---|
| | β | CI | p | β | CI | p | β | CI | p |
| Intercept | -0.045 | -0.078 – -0.012 | **0.007** | 0.065 | -0.010 – 0.140 | 0.087 | 0.062 | -0.057 – 0.182 | 0.306 |
| Time Awake (hours) | 0.031 | 0.024 – 0.039 | **<0.001** | 0.017 | 0.005 – 0.029 | **0.007** | 0.016 | -0.006 – 0.038 | 0.157 |
| Time Awake Squared (hours^2) | -0.002 | -0.003 – -0.002 | **<0.001** | -0.001 | -0.002 – -0.001 | **<0.001** | -0.001 | -0.002 – 0.000 | 0.053 |
| Age | | | | 0.002 | -0.006 – 0.009 | 0.635 | -0.014 | -0.054 – 0.026 | 0.492 |
| Sex [Female] | | | | -0.025 | -0.043 – -0.006 | **0.009** | -0.063 | -0.135 – 0.010 | 0.091 |
| Time of Day (Sine) | | | | -0.045 | -0.074 – -0.016 | **0.002** | -0.046 | -0.073 – -0.019 | **0.001** |
| Time of Day (Cosine) | | | | -0.084 | -0.107 – -0.062 | **<0.001** | -0.08 | -0.103 – -0.057 | **<0.001** |
| Total Typing Duration (minutes) | | | | -0.058 | -0.071 – -0.045 | **<0.001** | -0.058 | -0.071 – -0.045 | **<0.001** |
| Total Typing Duration Squared (minutes^2) | | | | 0.003 | 0.002 – 0.004 | **<0.001** | 0.003 | 0.002 – 0.004 | **<0.001** |
| Last Sleep Duration (minutes) | | | | 0 | -0.000 – 0.000 | 0.653 | 0 | -0.000 – 0.000 | 0.679 |
| Last Sleep Duration Squared (minutes^2) | | | | 0 | -0.000 – 0.000 | 0.271 | 0 | -0.000 – 0.000 | 0.436 |
| Weekend | | | | 0.008 | -0.014 – 0.030 | 0.464 | 0.009 | -0.013 – 0.031 | 0.420 |
| Device [Fitbit] | | | | -0.031 | -0.054 – -0.009 | **0.006** | -0.083 | -0.179 – 0.012 | 0.088 |
| Device [Garmin] | | | | -0.061 | -0.086 – -0.036 | **<0.001** | -0.185 | -0.295 – -0.075 | **0.001** |
| Keyboard Area (scaled) | | | | -0.007 | -0.012 – -0.001 | **0.012** | -0.007 | -0.012 – -0.002 | **0.010** |
| Rate of Deletion Zscore | | | | -0.367 | -0.382 – -0.351 | **<0.001** | -0.367 | -0.382 – -0.351 | **<0.001** |
| Time Awake x Age | | | | | | | 0.005 | -0.004 – 0.014 | 0.256 |
| Time Awake x Sex[Female] | | | | | | | 0.007 | -0.008 – 0.022 | 0.388 |
| Time Awake x Device[Fitbit] | | | | | | | 0.006 | -0.015 – 0.026 | 0.575 |
| Time Awake x Device[Garmin] | | | | | | | 0.021 | -0.004 – 0.045 | 0.095 |
| Time Awake x Last Sleep Duration | | | | | | | 0 | -0.000 – 0.000 | 0.587 |
| Time Awake Squared x Age | | | | | | | 0 | -0.001 – 0.000 | 0.200 |
| Time Awake Squared x Sex[Female] | | | | | | | 0 | -0.001 – 0.001 | 0.563 |
| Time Awake Squared x Device[Fitbit] | | | | | | | 0 | -0.001 – 0.001 | 0.970 |
| Time Awake Squared x Device[Garmin] | | | | | | | -0.001 | -0.002 – 0.001 | 0.327 |
| Time Awake Squared x Last Sleep Duration | | | | | | | 0 | -0.000 – 0.000 | 0.907 |
| Schwartz-type penalized Gaussian Pseudo-likelihood Criterion (SGPC); lower the better | 127152 | | | 120029 | | | 120065 | | |

There were different individual patterns of the change of typing speed over the course of time awake, as the two examples illustrated in **Fig 4**. Among several factors including age, sex, the type of wearable device used, and the duration of the most recent sleep episode, no covariate demonstrated a significant moderating effect on the relationship between time awake and typing speed (**Table 2, right column**).

## The relationship between time awake and rate of deletion

The adjusted model (2) had the best data fit for the relationship between time awake and rate of deletion according to SGPC. Based on model (2). After adjusting for covariates, the rate of deletion was significantly associated with both the

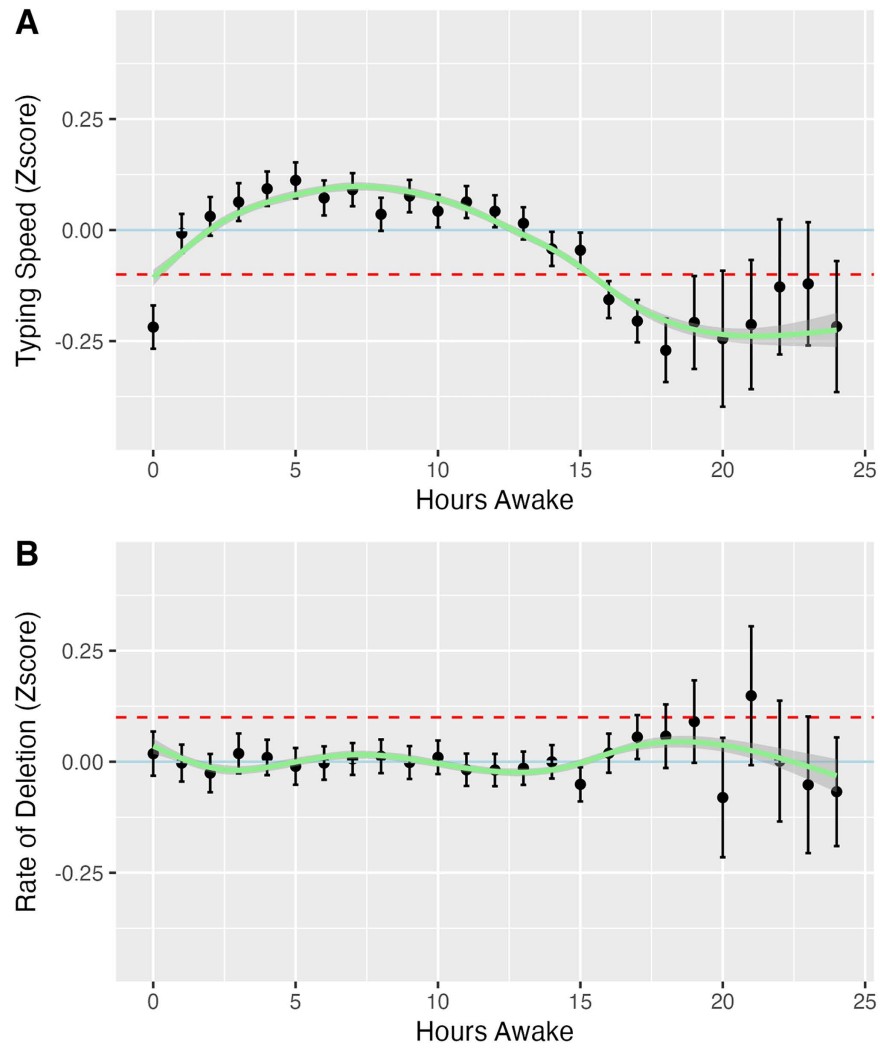

**Fig 3. Average typing speed (A) and average rate of deletions (B) by number of hours awake.** Typing speed and rate of deletions were converted to individual z-scores. The error bars represent 95%CI. The GAM-smoothed prediction lines from covariate-adjusted model (4) were plotted (green). The Blue solid line represents the individual average (z-score = 0), while the red dashed line represents -0.1 standard deviation in **(A)** and +0.1 standard deviation in **(B)**.

linear and quadratic terms of time awake (**Table 3, middle column**). Despite this significant association, the change in deletion rate was minimal, with none of the predicted rates of deletion exceeding 0.1 standard deviations above the average (**Fig 3B**). S3 Fig illustrates two examples of individual patterns showing different trends in the rate of deletion over time awake. We did not find any significant moderating effect of age, sex, the type of wearable device used, and the duration of the most recent sleep episode, on the relationship between time awake and rate of deletion (**Table 3, right column**).

## Diurnal rhythmicity

As shown above, time of day has a significant effect on tying speed (**Table 2**) but not rate of deletion (**Table 3**). After adjusting for covariates, typing speed exhibits significant diurnal rhythmicity with an amplitude of 0.10 (SE = 0.01), and the

PLOS Digital Health

**A**

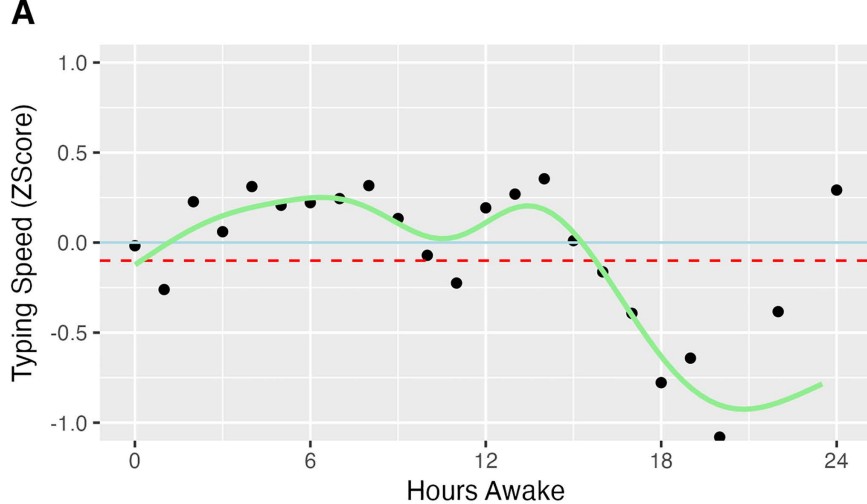

**B**

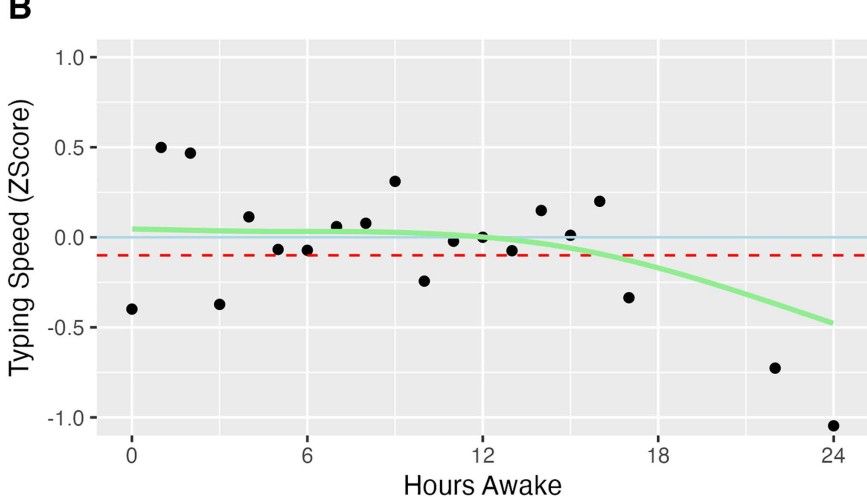

**Fig 4. Average typing speed by hours awake for two example subjects.** Typing speeds were converted to individual z-scores. The Blue solid line represents the individual average (z-score＝0), while the red dashed line represents -0.1 standard deviation below the individual average.

peak of the rhythm occurs at 11:49am. Across all four strata of different lengths of time awake, typing speed is consistently higher during the day compared to the night, with overall lower typing speed in longer time awake strata.(**Fig 5**)

### Sensitivity analyses: Restriction to a maximum of 18 hours awake

There were 43,413 (96.4% of all 45,042) recorded typing sessions from 365 subjects (99.7% of all 366) that occurred within 18 hours awake. Analyses conducted on this subset yielded similar results (S4 Fig).

### Sensitivity analyses: The effect of input languages

There were 40,834 (90.7% of all 45,042) recorded typing sessions from 363 subjects (99.2% of all 366) having the input mode of US English. Analyses conducted on this subset yielded similar results (S5 Fig).

**Table 3. Relationship between time awake and typing accuracy (rate of deletion). Left: unadjusted - model (1); Middle: adjusted - model (2); Right: adjusted with interactions - model (3)).**

| *Predictors* | Rate of deletion Zscore | | | Rate of deletion Zscore | | | Rate of deletion Zscore | | |
|---|---|---|---|---|---|---|---|---|---|
| | *β* | *CI* | *p* | *β* | *CI* | *p* | *β* | *CI* | *p* |
| Intercept | 0.01 | -0.016 – 0.036 | 0.463 | -0.084 | -0.157 – -0.010 | **0.025** | -0.101 | -0.220 – 0.017 | 0.094 |
| Time Awake (hours) | -0.003 | -0.010 – 0.003 | 0.291 | 0.015 | 0.007 – 0.023 | **<0.001** | 0.019 | -0.004 – 0.042 | 0.104 |
| Time Awake Squared (hours^2) | 0 | -0.000 – 0.001 | 0.241 | -0.001 | -0.001 – -0.000 | **<0.001** | -0.001 | -0.002 – 0.000 | 0.071 |
| Age | | | | -0.001 | -0.003 – 0.001 | 0.413 | -0.003 | -0.032 – 0.025 | 0.818 |
| Sex [Female] | | | | 0.004 | -0.001 – 0.010 | 0.117 | -0.014 | -0.064 – 0.036 | 0.587 |
| Time of Day (Sine) | | | | 0.019 | -0.003 – 0.042 | 0.092 | 0.02 | -0.004 – 0.045 | 0.098 |
| Time of Day (Cosine) | | | | -0.014 | -0.031 – 0.003 | 0.103 | -0.011 | -0.029 – 0.006 | 0.200 |
| Total Typing Duration (minutes) | | | | 0.013 | 0.002 – 0.024 | **0.022** | 0.013 | 0.002 – 0.024 | **0.023** |
| Total Typing Duration Squared (minutes^2) | | | | 0 | -0.001 – 0.001 | 0.737 | 0 | -0.001 – 0.001 | 0.733 |
| Last Sleep Duration (minutes) | | | | 0 | -0.000 – 0.000 | 0.737 | 0 | -0.000 – 0.001 | 0.495 |
| Last Sleep Duration Squared (minutes^2) | | | | 0 | -0.000 – 0.000 | 0.565 | 0 | -0.000 – 0.000 | 0.457 |
| Weekend | | | | 0.038 | 0.016 – 0.060 | **0.001** | 0.039 | 0.017 – 0.061 | **0.001** |
| Device [Fitbit] | | | | 0.009 | -0.000 – 0.019 | 0.058 | 0.018 | -0.044 – 0.079 | 0.572 |
| Device [Garmin] | | | | 0.003 | -0.013 – 0.019 | 0.710 | -0.025 | -0.126 – 0.076 | 0.630 |
| Keyboard Area (scaled) | | | | -0.001 | -0.004 – 0.001 | 0.341 | -0.001 | -0.004 – 0.002 | 0.445 |
| Typing Speed Zscore | | | | -0.372 | -0.388 – -0.356 | **<0.001** | -0.372 | -0.388 – -0.357 | **<0.001** |
| Time Awake x Age | | | | | | | 0 | -0.007 – 0.007 | 0.950 |
| Time Awake x Sex[Female] | | | | | | | 0.002 | -0.010 – 0.014 | 0.723 |
| Time Awake x Device[Fitbit] | | | | | | | -0.007 | -0.022 – 0.007 | 0.328 |
| Time Awake x Device[Garmin] | | | | | | | -0.001 | -0.022 – 0.020 | 0.936 |
| Time Awake x Last Sleep Duration | | | | | | | 0 | -0.000 – 0.000 | 0.740 |
| Time Awake Squared x Age | | | | | | | 0 | -0.000 – 0.000 | 0.823 |
| Time Awake Squared x Sex[Female] | | | | | | | 0 | -0.001 – 0.001 | 0.929 |
| Time Awake Squared x Device[Fitbit] | | | | | | | 0.001 | -0.000 – 0.001 | 0.158 |
| Time Awake Squared x Device[Garmin] | | | | | | | 0 | -0.001 – 0.001 | 0.532 |
| Time Awake Squared x Last Sleep Duration | | | | | | | 0 | -0.000 – 0.000 | 0.878 |
| SGPC; lower the better | 127472 | | | 120730 | | | 120781 | | |

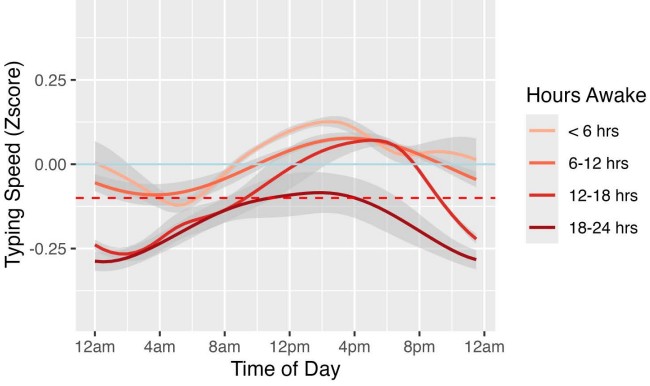

**Fig 5. The relationship between time of the day with typing speed stratified by time awake.** Dataset was split into 4 subgroups: <6 hours, 6-12 hours, 12-18 hours, and 18-24 hours. Each subgroup was fitted with model (4) respectively. The GAM-smoothed prediction lines from each model were plotted.

### Sensitivity analyses: The effect of wearable device types

Among 45,042 typing sessions included in the analysis, there were 32,940, 9,111, and 2,991 typing sessions linked with a sleep episode recorded by a Fitbit, Apple Watch or Garmin watch, respectively. No significant interaction was found between the device types and time awake on both of the typing performances (**Tables 2 and 3**). In addition, sensitivity analyses that repeated the same analysis within three device type subgroups, which contain typing sessions linked with only Fitbit-measured, Apple Watch-measured, or Garmin-measured sleep episodes, showed similar patterns with the results including all data (S6, S7 Figs).

### Sensitivity analyses: The effect of baseline sleep duration

There were 320 participants with at least one day of sleep data at baseline, contributing 41,048 typing episodes in total. Baseline sleep duration had no significant effect on either typing speed ($\beta = 1.8 \times 10^{-4}$, p = 0.59) or rate of deletion ($\beta = 3.2 \times 10^{-4}$, p = 0.19). There was also no significant interaction between baseline sleep duration and time awake (linear and quadratic terms) on typing speed ($\beta_{linear} = -8.1 \times 10^{-5}$, p = 0.24; $\beta_{quadratic} = 3.7 \times 10^{-6}$, p = 0.27) or rate of deletion ($\beta_{linear} = -6.0 \times 10^{-5}$, p = 0.26; $\beta_{quadratic} = 2.2 \times 10^{-6}$, p = 0.36).

## Discussion

In this study, we investigated the potential utility of real-time, unobtrusive and ambulatory smartphone keyboard typing performance metrics to continuously and passively measure mental fatigue signals. Using over 45,000 typing episodes assessed through SensorKit and sleep information recorded by wearable devices, among over 350 medical interns, we found that typing speed was significantly associated with time awake, after adjusting for its diurnal rhythmicity, and other confounding factors. Specifically, we found that the typing speed increased upon waking and peaked at 0.098 SD above average 7.5 hours after wake-up and then declined to over 0.1 SD below average by 15.3 hours awake. These results remained robust after considering the effect of input languages and the type of wearable devices measuring sleep time.

Traditional tools of assessing alertness and fatigue, such as the PVT, that require active participation for assessment, have limited ability to continuously detect mental fatigue in the real world. Our findings suggest typing performance, especially smartphone typing speed, may be a promising new tool to passively and continuously follow mental fatigue in a naturalistic setting. Monitoring smartphone typing speed takes advantage of the ubiquitous use of mobile phones, imposes virtually no burden on the users while protecting user privacy. This tool could be valuable in real-word settings such as ensuring the work safety and quality of shift workers who are prone to fatigue-related errors. It could also be applied in areas where continuous alertness is essential, such as healthcare, manufacturing, public safety and transportation, to provide real-time feedback and inform interventions such as scheduling adjustments, timely breaks, and targeted wellness programs that enhance worker safety and operational efficiency. The tool could also be useful for individuals, providing information on when they should focus on cognitively demanding tasks and when they should disengage.

Our findings from typing speed are consistent with findings from active assessments of mental fatigue using tools such as the PVT. For example, the critical wake period beyond which PVT response lapse increased was estimated to be 15.8 hours [21], aligning closely with our observation regarding the amount of time awake (15.3 hours) when typing speed reduced to 0.1SD below average. Yet it is also important to note that our analyses identified periods of elevated typing speeds suggesting above-average alertness earlier in the day (~7–8 hrs after awakening). Future research directly comparing PVT and typing speed within the same group of subjects would help to further demonstrate the validity of using typing speed as a real-world and passively-sensed proxy for PVT.

The significant associations between typing speed with both time awake and time of day is consistent with the well-described two-process model of sleep [30]. This model describes the interaction of two distinct processes to produce consolidated periods of wakefulness and sleep: the homeostatic process (Process S) drives sleep need based on prior sleep and wakefulness, and the central circadian pacemaker (Process C) times the propensity for alertness

to align with the daylight period. In a normally entrained, non-sleep deprived individual, upon awakening the homeostatic sleep drive is low but accumulates throughout the day with sustained wakefulness. The circadian alerting signal increases in opposition to the growing homeostatic sleep drive such that wakefulness can be sustained across the daylight period until bedtime. Approximately two hours prior to habitual sleep onset, the circadian alerting signal drops and in conjunction with an elevated homeostatic sleep drive, promotes sleep onset. Our findings align with these processes, with an initial increase in typing speed commensurate with an increasing circadian alerting signal (process C) followed by a decline in typing speed that aligns with an accumulation of the homeostatic sleep drive (process S) and drop in the circadian alerting signal. Further research exploring these dynamics may provide better predictions of cognitive performance and fatigue.

In addition to our findings related to the average trend among participants, we also found that there are individual differences in the relationship between typing speed and time awake. As illustrated in **Fig 4**, individuals differed in the timing and shape of their typing speed peaks and troughs with reference to time awake. Furthermore, meaningful departures from personal past typing speed trends can help to identify additional important driving factors. Specifically, continuous data collection enables establishment of personalized models that can be interrogated for associations with other individual differences (e.g., genotype, chronotype, type of work) and within-person predictors (e.g., exertion, breaks). Based on the results of the interaction tests, basic demographics like age and sex, as well as baseline and previous sleep duration, did not seem to be the factors moderating the relationship between time awake and typing speed. Future work is warranted in identifying between and within individual characteristics that may signal differences in mental fatigue development. From an applied perspective, tailored models could be developed to anticipate potential critical periods that maximize alertness and minimize fatigue for use in personalized feedback and intervention. Ongoing individualized monitoring of typing performance trends could also serve as a sentinel for unexpected critical drops in alertness.

In contrast to typing speed, typing accuracy, represented by the rate of deletion, remained relatively stable with time awake and time of day. This suggests that rate of deletion may not be as sensitive a measure of mental fatigue under the conditions studied. One possible explanation is that deletions during typing can reflect not just mistakes, but ongoing, deliberate editing, such as when composing charts or work emails, thus limiting its usefulness as a fatigue indicator. Additionally, autocorrection features on devices may further confound accuracy measures. Therefore, while typing accuracy seems less sensitive to time awake in this context, it warrants further exploration.

This study has several strengths. First, the study benefits from high statistical power with longitudinal data collected over two months from more than 350 subjects, resulting in over 45,000 typing episodes, a much larger sample than previous studies on typing performance and mental fatigue signals [5–8, 12–15]. Second, the study population, medical interns, typically endure long work hours and follow shift work schedules, which ensures that the collected data are well-distributed across various times of the day and durations of wakefulness. Additionally, interns frequently engage in typing on their phones for both professional and personal purposes. This group is also susceptible to fatigue-related work errors. These characteristics make medical interns an important and ideal group to study mental fatigue and its related markers. Third, with SensorKit and wearable technology, we were able to objectively collect both typing and sleep data in a naturalistic setting. This approach provides better accuracy than self-reported data, such as sleep diaries; and places minimal burden on subjects compared to assigning specific typing tasks. The utilization of SensorKit also allows massive scalability of the tool for mental fatigue assessment, if future studies confirm and extend findings from the study.

There are also some limitations to this study. First, there may be other confounding factors, such as caffeine and nicotine intake, medication, medical history [31], workload and demands, as well as motivation and context of typing, that could influence the associations between typing performance and time awake. For example, a higher workload may cause fatigue to accumulate more rapidly after shorter periods awake, while caffeine consumption may increase

alertness even after extended wakefulness. Additionally, typing speed may differ between texting with friends and work-related emailing that requires greater cognitive effort. In fact, even during the periods associated with the highest average typing speed, there are still >40% typing sessions with individual typing speed z-scores below -0.1 (S8 Fig), reflecting meaningful variability that may be driven by these contextual factors. However, these factors were not measured and thus not accounted for in our study. Future studies could benefit from systematically examining these contextual influences. Second, the study sample was limited to young, healthy adults engaged in medical internships. Conducting additional research with diverse population groups will help to better determine the generalizability of this innovative tool. Third, due to device-wearing habits and the limited battery life, there was missingness in sleep data collection. To mitigate this issue, we restricted the time interval between the previous wake and the subsequent typing session to 24 hours or less. This approach helps prevent inaccurately prolonged wake periods linked with typing sessions due to unrecorded sleep episodes. Finally, this study did not take into account the time spent at work, which could be another factor contributing to mental fatigue. Future analyses, incorporating other SensorKit metrics such as location types, may provide deeper insight into this aspect.

In summary, our findings highlight the promise of using typing performance, particularly typing speed, as a novel and practical method for unobtrusive, ambulatory mental fatigue assessment in the real world. This approach, utilizing common smartphone technology, has the potential to provide valuable insights into cognitive alertness and fatigue and to offer a scalable low-cost, low-burden and privacy-preserving tool that can be embedded organically in everyday life. Future studies should aim to further validate and refine this approach across diverse populations and settings, as well as explore and evaluate personalized intervention strategies.

## Supporting information

**S1 Fig. SensorKit Keyboard Usage data collection.**
(DOCX)

**S2 Fig. The numbers of: typing sessions by hours awake (A), subjects by hours awake (B), typing sessions by typing speed (C), and typing sessions by rate of deletion (D) in the sample.**
(DOCX)

**S3 Fig. Average rate of deletion of two example subjects by hours awake.**
(DOCX)

**S4 Fig. Average typing speed (A) and average rate of deletions (B) by number of hours awake, limited to a maximum of 18 hours awake.**
(DOCX)

**S5 Fig. Average typing speed (A) and average rate of deletions (B) by number of hours awake, US English input mode only.**
(DOCX)

**S6 Fig. Average typing speed versus hours awake, by the type of wearable device recording sleep.**
(DOCX)

**S7 Fig. Average rate of deletion versus hours awake, by the type of wearable device recording sleep.**
(DOCX)

**S8 Fig. Proportion of typing sessions with low speed by hours awake in the sample.**
(DOCX)

## Author contributions

**Conceptualization:** Yu Fang, Srijan Sen, Zhenke Wu.

**Data curation:** Yu Fang.

**Formal analysis:** Yu Fang, Peter Yang, Zhenke Wu.

**Funding acquisition:** Srijan Sen.

**Investigation:** Yu Fang, Cathy Goldstein, Aidan G.C. Wright, Amy S.B. Bohnert, Srijan Sen, Zhenke Wu.

**Methodology:** Yu Fang, Peter Yang, Zhenke Wu.

**Project administration:** Elena Frank.

**Resources:** Elena Frank, Cathy Goldstein, Vik Kheterpal, Srijan Sen.

**Software:** Yu Fang, Peter Yang, Vik Kheterpal, Zhenke Wu.

**Supervision:** Srijan Sen, Zhenke Wu.

**Validation:** Yu Fang, Peter Yang, Zhenke Wu.

**Visualization:** Yu Fang, Peter Yang.

**Writing – original draft:** Yu Fang.

**Writing – review & editing:** Yu Fang, Peter Yang, Elena Frank, Cathy Goldstein, Aidan G.C. Wright, Amy S.B. Bohnert, Vik Kheterpal, Srijan Sen, Zhenke Wu.

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
