## [Decision Letter · Decision Letter 0]

8 Dec 2025

Response to Reviewers
Revised Manuscript with Track Changes
Manuscript
**Journal Requirements:**

1. We have amended your Competing Interest statement to comply with journal style. We kindly ask that you double check the statement and let us know if anything is incorrect.

2. Please provided your detailed Financial Disclosure statement. This is published with the article. It must therefore be completed in full sentences and contain the exact wording you wish to be published.

1. Please clarify all sources of funding (financial or material support) for your study. List the grants (with grant number) or organizations (with url) that supported your study, including funding received from your institution.

2. State the initials, alongside each funding source, of each author to receive each grant.

3. State what role the funders took in the study. If the funders had no role in your study, please state: “The funders had no role in study design, data collection and analysis, decision to publish, or preparation of the manuscript.”

4. If any authors received a salary from any of your funders, please state which authors and which funders.

3. Please provide separate figure files in .tif or .eps format.

4. Please insert an Ethics Statement at the beginning of your Methods section, under a subheading 'Ethics Statement'. It must include:

1) The name(s) of the Institutional Review Board(s) or Ethics Committee(s)

2) The approval number(s), or a statement that approval was granted by the named board(s)

3) (for human participants/donors) - A statement that formal consent was obtained (must state whether verbal/written) OR the reason consent was not obtained (e.g. anonymity). NOTE: If child participants, the statement must declare that formal consent was obtained from the parent/guardian.

5. We have noticed that you have uploaded Supporting Information files, but you have not included a list of legends. Please add a full list of legends for your Supporting Information files after the references list.

**Reviewers' Comments:**

**Comments to the Author**

1. Does this manuscript meet PLOS Digital Health’s publication criteria?

Reviewer #1: Yes

Reviewer #2: Yes

Reviewer #3: Yes

Reviewer #4: Yes

Reviewer #5: Yes

2. Has the statistical analysis been performed appropriately and rigorously?

Reviewer #1: Yes

Reviewer #2: Yes

Reviewer #3: Yes

Reviewer #4: Yes

Reviewer #5: Yes

3. Have the authors made all data underlying the findings in their manuscript fully available (please refer to the Data Availability Statement at the start of the manuscript PDF file)?

Reviewer #1: Yes

Reviewer #2: Yes

Reviewer #3: Yes

Reviewer #4: No

Reviewer #5: No

4. Is the manuscript presented in an intelligible fashion and written in standard English?

Reviewer #1: Yes

Reviewer #2: Yes

Reviewer #3: Yes

Reviewer #4: Yes

Reviewer #5: Yes

Reviewer #1: Dear Authors,

Thank you for your submission of the manuscript 'Patterns of smartphone typing performance by time awake: implications for unobtrusive ambulatory mental fatigue assessment.' The work is very interesting and depicts a novel form of assessing mental fatigue through use of smartphone keystrokes. The methods and results are very detailed. The only minor revision is that there are some papers cited in the references which I cannot find cited in the text (27-30) which are relevant articles that I would have liked to see discussed either in the introduction's review of related works, or in the discussion.

Reviewer #2: This paper benefits from several strengths, thereby making is complete and sound:

- The research question is very appropriate and interesting to address in a timely fashion

- The examined cohort is extensive and observed over a significant portion of time

- The statistical analysis is sound and particularly smart being based on a generalized estimating equation. This method is particularly appropriate for examining the relationship between typing and awake time since this equation typically estimates the parameters of a generalized linear model with a possible unmeasured correlation between observations from different time points, which is the case here

- The relationships between the independent and dependent variables are properly analyzed and discussed.

Tables 2 and 3 are challenging to browse in the submission as their indentation is fragmented. This should be improved in the final version with mid lines. When there is a significant p-value, maybe compute the effect size to determine the magnitude of the effect, which might be interesting since the sampling is extensive.

Reviewer #3: Strengths

This is an interesting and timely study that investigates the association between smartphone typing performance and time awake, with the broader goal of exploring whether typing behavior may serve as a passive indicator of mental fatigue. The scale of data collection is impressive—over 45,000 typing sessions from 366 medical interns—and demonstrates substantial effort and rigorous data handling. The statistical approach, including generalized estimating equations, GAM smoothing, and cosinor analysis, is methodologically sound and appropriate for modeling repeated measures in a real-world setting. The manuscript is clearly written, and the overall analytical framework is well-structured. This line of work has strong potential for digital-health applications, particularly in fatigue-sensitive occupations, though several aspects would benefit from clarification to ensure accurate interpretation of the findings.

Weaknesses and suggestions

1. While the study leverages “time awake” as an objective proxy for mental fatigue, it is important to note that time awake cannot fully replace or directly represent mental fatigue. Although the relationship between prolonged wakefulness and reduced alertness is well documented in prior literature, the study does not include any direct measurement of subjective or objective fatigue. Therefore, language implying fatigue detection, causal inference, or diagnostic capability should be avoided. The findings should be framed strictly as associations between typing performance and time awake.

2. Regarding typing behavior, several potentially important confounding factors were not included in the analysis. Although age, sex, device type, and time of day were appropriately adjusted for, medical interns’ typing performance is likely influenced by situational elements such as clinical workload, communication context, or multitasking demands. These real-world factors may partly explain why many prior studies were conducted in controlled laboratory environments, where such variability can be minimized. Although these contextual influences are difficult to capture in passive sensing data, they may meaningfully affect both typing speed and accuracy. A more explicit discussion of how unmeasured contextual factors could shape the observed associations would strengthen the interpretation of typing performance as a potential fatigue indicator. In addition, presenting the frequency of low typing-performance z-score sessions occurring during periods when individuals typically perform well may help readers assess whether these sessions reflect meaningful variability or context-driven outliers.

3. The manuscript models both circadian timing and time awake, but partial collinearity between these variables may still concern readers—especially given that participants exhibit similar wake-up times (around 6 AM). The authors could strengthen the analysis by briefly reporting the correlation between time awake and time of day in this dataset, and by presenting a partial residual plot (from a GAM or GEE model) showing typing speed as a function of time awake after adjusting for time of day.

4. Finally, the interpretation of the deletion rate deserves clarification. Although statistically significant, the deletion rate shows minimal practical variation and appears much less sensitive to time awake than typing speed. Providing a concise explanation in the Results or Discussion—such as its low baseline variability, contextual dependence, or inherently smaller dynamic range—would make the rationale for focusing primarily on typing speed clearer.

Reviewer #4: Overall, the manuscript contributes valuable and novel insights with methodological rigor, but the authors should address and highlight data sharing limitations in accordance with PLOS policies. This will support the journal’s commitment to open science while respecting data privacy and third-party restrictions.

Reviewer #5: The paper presents a study investigating the relationships between fatigue (measured as time from the last sleep episode) and speed and accuracy of typing on a smartphone in a large sample of MD interns, a population well-suited for a study of fatigue. The results show that typing speed and accuracy increase for the first 7 hours of waking time and decrease from hour 7 onwards, with speed of typing after 15 hours of waking time being more than 1 standard deviation lower than an individual average. The analysis of time of day and clarification of its additive effect to that of sleep duration is important for supporting the general conclusion and well-presented in Figure 5. This is a timely and insightful study and a well-written paper, I wholeheartedly support its publication as is.

Suggestions for improvement:

- Hours awake after about 18 yield very low number of typing instances, and if all the instances are coming from the same few participants this could skew the results and the conclusions. An additional sensitivity analysis could limit the hours awake to 18 (inclusive). An additional plot could show the number of participants with at least one typing session for each hour (like Figure S2A, but with count of participants instead of events on the Y axis).

- In the formulae for computing Z-scores, I am not sure if the last subscript t (after the curly brackets) is needed, and if needed, shouldn’t it be t (not s) for the denominator in the deletion rate formula?

- For the typing statistics, was any typing included or only messaging and notes? E.g., typing in the browser, phone search, calendar, etc? This information could be included for clarity.

- Inclusion of both sine and cosine for the time of day could be explained in a bit more detail. Why would both be included?

**Do you want your identity to be public for this peer review?** For information about this choice, including consent withdrawal, please see our Privacy Policy

Reviewer #1: No

Reviewer #2: No

Reviewer #3: No

Reviewer #4: **Yes:** Dr S Suryanarayana Raju D

Reviewer #5: No

**Figure resubmission:**

**Reproducibility:** To enhance the reproducibility of your results, we recommend that authors of applicable studies deposit laboratory protocols in protocols.io, where a protocol can be assigned its own identifier (DOI) such that it can be cited independently in the future. Additionally, PLOS ONE offers an option to publish peer-reviewed clinical study protocols. Read more information on sharing protocols at https://plos.org/protocols?utm_medium=editorial-email&utm_source=authorletters&utm_campaign=protocols

---

## [Decision Letter · Decision Letter 1]

18 Feb 2026

Patterns of smartphone typing performance by time awake: implications for unobtrusive ambulatory mental fatigue assessment

PDIG-D-25-00755R1

Dear Ms Fang,

We are pleased to inform you that your manuscript 'Patterns of smartphone typing performance by time awake: implications for unobtrusive ambulatory mental fatigue assessment' has been provisionally accepted for publication in PLOS Digital Health.

Best regards,

Louise A C Millard, PhD

Section Editor

PLOS Digital Health

**Additional Editor Comments (if provided):**

**Reviewer Comments (if any, and for reference):**

Reviewer's Responses to Questions

**Comments to the Author**

Reviewer #3: All comments have been addressed

Reviewer #5: All comments have been addressed

publication criteria?

Reviewer #3: Yes

Reviewer #5: Yes

3. Has the statistical analysis been performed appropriately and rigorously?

Reviewer #3: Yes

Reviewer #5: Yes

4. Have the authors made all data underlying the findings in their manuscript fully available (please refer to the Data Availability Statement at the start of the manuscript PDF file)?

Reviewer #3: Yes

Reviewer #5: Yes

5. Is the manuscript presented in an intelligible fashion and written in standard English?

Reviewer #3: Yes

Reviewer #5: Yes

Reviewer #3: All of my questions and suggestions have been properly addressed by the author. I don't have any other questions.

Reviewer #5: The authors addressed all my comments and I have no further suggestions.

**Do you want your identity to be public for this peer review?** For information about this choice, including consent withdrawal, please see our Privacy Policy

Reviewer #3: No

Reviewer #5: **Yes:** Veronica Dudarev
